# Myofascial Treatment for Microcirculation in Patients with Postural Neck and Shoulder Pain

**DOI:** 10.3390/diagnostics11122226

**Published:** 2021-11-29

**Authors:** Jian-Guo Bau, Shyi-Kuen Wu, Bo-Wen Huang, Tony Tung-Liang Lin, Shih-Chung Huang

**Affiliations:** 1Department of Biomedical Engineering, Hungkuang University, Taichung City 433, Taiwan; baujg@sunrise.hk.edu.tw (J.-G.B.); wosooz536@gmail.com (B.-W.H.); 2Department of Physical Therapy, Hungkuang University, Taichung City 433, Taiwan; 3Department of Physical Medicine and Rehabilitation, Taichung Veterans General Hospital, Taichung City 407, Taiwan; tllin@vghtc.gov.tw; 4Division of Cardiology, Kuang-Tien General Hospital, Taichung City 433, Taiwan

**Keywords:** postural neck pain, myofascial therapy, microcirculation, laser-doppler flowmetry

## Abstract

Vascular impairment is a crucial factor associated with chronic muscle pain, but relevant research from the microcirculatory aspect is lacking. Here, we investigated the differences in neck muscle microcirculation detected through laser-doppler flowmetry (LDF) and cervical biomechanics by a videofluoroscopic image in asymptomatic participants and patients with postural neck and shoulder pain. To understand the mechanism behind the effect of myofascial treatment, transverse friction massage (TFM) was applied and the immediate effects of muscular intervention on microcirculation were monitored. In total, 16 asymptomatic participants and 22 patients (mean age = 26.3 ± 2.4 and 25.4 ± 3.2 years, respectively) were recruited. Their neck muscle microcirculation and spinal image sequence were assessed. The differences in the baseline blood flow between the asymptomatic and patient groups were nonsignificant. However, the standard deviations in the measurements of the upper trapezius muscle in the patients were significantly larger (*p* < 0.05). Regarding the TFM-induced responses of skin microcirculation, the blood flow ratio was significantly higher in the patients than in the asymptomatic participants (*p* < 0.05). In conclusion, postintervention hyperemia determined through noninvasive LDF may be an indicator for the understanding of the mechanism underlying massage therapies and the design of interventions for postural pain.

## 1. Introduction

In a work environment, people frequently complain about neck, shoulder, and back problems, especially when using a computer or smartphone while in an inappropriate posture for a long period. In such a position, the head moves forward, which increases the weight of the head supported by the neck muscles, and this may lead to pathological changes in the head and neck regions [1]. This musculoskeletal disorder is often characterized by pain, muscle weakness, and restricted range of motion (ROM). Work-related neck and upper-limb disorders have been associated with the long-term demands of computer work and prolonged holding of a static posture [1,2]. The static posture of the neck and shoulders has been frequently assumed to be a possible risk factor for work-related neck and upper-limb disorders. Evidence suggests that the prolonged static posture is connected to increased muscle loading and subsequent development of the symptoms in the neck and upper body [2,3]. In addition to the static posture, the nature of the repetitive work tasks may affect the movement patterns in the neck and shoulder regions. The physical requirements at the workplace, e.g., work requiring prolonged static posture and highly repetitive work may be some of the risk factors for increased musculoskeletal disorders over the neck and shoulder areas [1].

In physical medicine, good posture refers to the proper alignment of the body segments between the postural extremes. Excessive or reduced spinal curvature is considered deviation from the neutral alignment. In an ideal posture, the proper alignment of the bony segments requires the least amount of energy to maintain a balanced position [3,4,5,6,7,8]. Long-term improper static posture typically occurs in the neck and shoulder regions and can increase static muscle tension in the upper body. Excessive forward neck flexion may lead to increased muscular tension in the postural muscles along with increased compressive forces in the facet articulations of the cervical spine. Sustained forward neck flexion of the cervical spine may increase the compressive loading in the cervical spine, and thus, a creep response occurs in the soft tissues of the neck [5]. This may increase neck and shoulder pain risk [3,4,5]. With an increase in the prevalence of a sedentary lifestyle, poor posture is characterized by the elevation and protraction of the shoulders (rounded shoulder), winging of the scapula, and protraction of the head [4]. This atypical posture results in the overstress of the occipital and cervical junction and the altered motion of the neck and shoulder joints. The patients presenting with neck and shoulder pain frequently demonstrate such a posture in clinical observation and radiological images [3,4,5]. In clinical research, some researchers have reported significant differences in the relaxed or resting head and neck postures between healthy people and symptomatic patients [6,7,8,9]. Although goniometers or inclinometers are usually used to measure the cervical range of motion (ROM) for the evaluation in patients with neck pain [10,11], large variations in spinal posture and motion have been noted among individuals [6,7]. The spinal flexibility tests—including trunk rotation, trunk later bending, and sit and reach tests—are indirect flexibility tests used for the clinical examination of joint ROM and can only provide information on whole-body flexibility. In addition, the obtained measurements are subject to systematic and random errors. Several factors, such as muscles, ligaments, tendons, skin, fat (or adipose) tissue, and joint structure must be considered when assessing flexibility around a joint. In contrast to the measurement of the global neck ROM that does not reveal the precise motion or restriction of individual vertebral segments, some researchers have applied videofluoroscopy to evaluate different motion patterns or anatomic changes between normal and pathological spine [12,13,14].

The cervical muscles are important for maintaining the stability and posture of the head and neck, and the sustained neck and shoulder posture that is frequently assumed by office workers, is a possible risk factor in work-related neck and upper limb disorders [1,2]. The increased compressive loading in the cervical spine and a creep response in the soft tissues have been suggested to lead to muscle dysfunction and neck pain syndromes. Clinical physical examination on muscular dysfunction reveals muscle asymmetry such as tender, taut, and palpable bands of muscle fibers, known as trigger points. A myofascial trigger point is usually associated with a painful restricted ROM and comprised circulation [15,16]. Soft tissue therapies are commonly applied in the treatment of postural neck pain and the related impairments [17,18,19,20]. Their results may be effective by decreasing pain, increasing ROM, and assisting the healing of spinal symptoms [17,21]. The transverse friction massage (TFM) is a technique applied perpendicular to the muscle fibers that promote optimal collagen healing by increasing circulation and decreasing collagen cross-linking. It has been found to reduce tenderness in the myofascial trigger points and to increase the ROM [19,20]. Although some patients with spinal issues demonstrate clinically meaningful improvements in pain relief and functional recovery after muscular interventions, the possible mechanism underlying the effects of muscular intervention on the intervertebral motion alternations warrants investigation.

Although circulatory impairment is usually acknowledged as an important contributing factor for acute or chronic muscle pain, little attention has been paid to the mechanism underlying postural neck and shoulder pain and clinical muscular intervention by the microcirculatory aspect. Laser-doppler flowmetry (LDF) is one of the most popular techniques applied to the assessment of tissue blood perfusion. In Strøm’s [22] study, intramuscular blood flux and muscle activity in the upper trapezius were studied through LDF and electromyography, respectively, and a positive correlation between pain and blood flux was found in people with chronic shoulder and neck pain. The vascular characteristic was found to be more sensitive than the muscle activity—recorded through noninvasive electromyography. However, since in the aforementioned and most of the related studies, the tissue perfusion signals were determined using the single-fiber LDF technique with an optic-fiber probe inserted invasively into the muscular tissue [23,24]; therefore, this technique may not be practical for regular clinical application. By contrast, Clough et al., developed a high-power LDF with a wide separation skin probe to explore its potential for noninvasive application when assessing the microcirculation of deeper tissues in humans [25]. As the muscular dysfunction may be related to the neck pain and lead to spinal motion restriction, evaluating the muscular microcirculation over the head and neck regions and its changes after myofascial interventions in subjects with different vertebral range of motion is crucial for understanding the intervertebral biomechanics. This may also be helpful for designing the clinical intervention on patients with postural neck pain.

Since both the symptoms of microcirculatory insufficiency [26] and the high-blood-flow [27] were found in patients with neck and shoulder pain in previous studies, our first hypothesis states that the standard deviation of the skin microcirculation around the neck and shoulders of patients with limited cervical spine movement, which was determined through videofluoroscopy, should be larger. In addition, a sustained hyperemia was observed in the shoulder muscles of patients with neck and shoulder pain after the intervention of the office works [22], our second hypothesis states that the effect of massage intervention should be more vigorous in patients with postural neck and shoulder pain. The microcirculatory characteristics of the patients and the effects of muscular intervention on microcirculation were investigated through noninvasive LDF in the studies.

## 2. Materials and Methods

### 2.1. Sample Size Calculation

The sample size was calculated through software GPower (version 3.1, Universität Kiel, Germany) [28] In order to study the differences in the effect of transverse friction massage on microcirculation between asymptomatic participants and patients with postural neck and shoulder pain, a small-scale study was conducted and we found the mean blood flow ratio of asymptomatic subjects and patients were 1.2 and 2.6 respectively; the standard deviation of the two groups were 0.5 and 2.0, respectively. According to the pilot data, at least 15 participants were required to achieve 95% statistical confidence, an 80% statistical power analysis (α = 0.05, β = 20%), and two-tailed test.

### 2.2. Subjects

In total, 16 healthy adults (8 men and women) without any neck symptoms within recent 4 weeks participated in this study. A participant was excluded if they (1) had a history of cervical trauma or surgery, (2) had bone pathology, (3) had an arthritic or other inflammatory disorder, (4) were pregnant, and (5) had restrictive muscle spasms. The participant age ranged from 20 to 50 years. Next, 22 age-matched patients (11 men and women) with neck and shoulder pain or symptoms such as pain, muscle spasm, and motor dysfunction within the recent 3 months were recruited. A patient was excluded if they (1) had a history of cervical surgery, such as disc replacement, bone fusion, and discectomy; (2) had significant potential for spinal cord injury, such as cord impingement from a large disk herniation; (3) had advanced cervical spondylosis; (4) had severe spinal stenosis; (5) had inflammatory arthritic disorders, including ankylosing spondylitis and rheumatoid arthritis; (6) had severe spinal instability; and (7) were pregnant. The clinical characteristics and Neck Disability Index (NDI) questionnaire responses of all the participants were documented. The NDI is a 10-item questionnaire about symptoms relevant to cervical spine pathology. The NDI consists of ten domains—pain intensity, personal care, lifting, reading, headache, concentration, work, driving, sleep, and recreation- designed to assess the level of disability in patients with neck pain [29].

The confirmation of the myofascial trigger points was performed using the five diagnostic criteria described by Simons et al. [15]: (1) presence of a palpable taut band in a skeletal muscle, (2) presence of a hypersensible tender spot in the taut band, (3) local twitch response elicited by the snapping palpation of the taut band, (4) reproduction of the typical referred pain pattern of the trigger point in response to compression, and (5) spontaneous presence of the typical referred pain pattern and/or patient recognition of the referred pain as familiar.

### 2.3. Microcirculatory Measurement Protocol and Data Analysis

Before data collection, the participants were asked to stay in the experimental environment with the temperature maintained at 26 ± 1 °C for at least 10 min. The participants adopted a sitting position with back support for watching a computer monitor. The microcirculation of upper trapezius, sternocleidomastoid (SCM), and masseter muscles was individually measured through LDF. For each muscle, a 210 s baseline was obtained in the relaxed state, and the muscles received the TFM for 2 min afterwards. Then, the other 210 s measurement was taken to monitor the effect of the muscular intervention. To correctly locate the position of the two measurements, the holder used to fix the probe was not removed during the massage. The skin probe of high-power LDF was placed on the midpoint between the cervical 7 (C7) vertebra and the acromion. The skin probe of the low-power LDF was placed lateral to that of the high-power one at a 1 cm distance. The measurements were conducted on both sides of the participants.

The microcirculatory flux of a muscle was simultaneously detected using high-power LDF (moorVMS-LDF1-HP; Moor Instruments, Devon, UK) and low-power LDF (moorVMS-LDF; Moor Instruments, Devon, UK). A noninvasive skin probe including two flexible glass optic fibers was used; of these fibers, one transmitted a 785 nm laser to the tissue surface, whereas the other collected the light scattered from the tissue and moving blood cells and transmitted back to LDF. If the light scattered from the moving blood cells, the light frequency shifted to a higher frequency, and the frequency difference between the incident and scattered light—which is called Doppler shift—is proportional to the velocity of the moving cells. High-power LDF along with a wide separation of transmitting and receiving fibers in the probe (VP1-V2-HP) enables a greater sampling depth and a larger volume to be monitored; this reduces site-to-site variations. With a wide separation of fibers (4.0 mm) for the high-power LDF versions, the sampling depth can be >1.4 mm [25], and the corresponding laser power to be used is 5–12 mW (maximum power = 20 mW, complying with Laser Safety Classification—Class 3R per IEC 60825-1:2007). By contrast, the sampling depth of the low-power LDF (typically 0.5–1.2 mW for standard systems) with optic-fiber separation of 0.5 mm for the probe (VP1T) is <1 mm. The sampling depth of high-power LDF is approximately where the arterioles in the dermis are located, where low-power LDF can only sample the capillaries in the epidermis. The probes used here were biocompatible for human use. The microcirculatory signals were acquired with a sampling frequency of 40 Hz and then sent to a personal computer with software provided by the manufacturer moorVMS-PC (version 3.1, Devon, UK) for further analysis.

To compare the effects of massage on microcirculation between the groups, the mean value of the post-intervention blood flow was divided by the individual baseline value to obtain the normalized blood flow—which was named blood flow ratio. Here, a ratio of >1 indicated that post-intervention blood flow was higher than the baseline value. Normalization could also eliminate the individual differences.

### 2.4. Videofluoroscopic Image Measurement and Data Analysis

For the neck movement imaging, a videofluoroscopy system for Toshiba (Tokyo, Japan)—comprising an X-ray generator that can operate at low milliamperage settings—was applied to evaluate cervical spine movement at a rate of 30 frames/s. An image intensifier as an image receptor was used for videofluoroscopy, and this permitted imaging with very low radiation exposure. The videofluoroscopy screen procedure lasted 10–20 s, which is equivalent to one or two typical chest X-ray radiation doses (0.06–0.12 mSv) but much lower than the natural background radiation dose of 2.0 mSv per year or a conventional chest CT of 2.0 mSv.

Before actual screening, the participants stood in front of an X-ray beam of videofluoroscopy and practiced active cervical flexion and extension movement three times with correction to reduce the trunk and out-of-plane motions. The flexion and extension movements were performed within 10 s. The recorded video images of spinal motion were then transformed into the sequences of bitmap images with the aid of the software program Canopus Edius (Canopus, Santa Ana, CA, USA). In total, 10 images in evenly divided intervals of each cervical motion range were selected for digitizing. During image processing, the positions of the 22 bony landmarks were digitized using SigmaScan (version 5.0; SPSS, Chicago, IL, USA) on a high-resolution monitor. The anatomical identification of the bony landmarks was based on the well-accepted radiographic method of Frobin et al. [30]. These were two inferior corners of the C2 vertebra and the right–left corners of the superior and inferior endplates from the C3 to the C7 vertebra in Figure 1. The methods for identifying vertebral landmarks were blinded between examiners and image sequences were digitized by two trained laboratory members. These vertebral landmarks are digitized three times each, and the mean values of the three measurements were used for subsequent analysis. The differences in intervertebral relationship and biomechanics were assessed using a videofluoroscopic imaging protocol in the asymptomatic participants and patients with postural neck and shoulder pain. The test–retest reliability of the digitizing procedures within raters was examined at 2 week intervals for four participants.

The MATLAB computer programs were written to construct the midplanes of the vertebrae defined as a line running through the midpoints between the anterior and posterior two corners, and the bisectrix between the two midplanes were derived [31]. The perpendiculars were constructed from the anterior–inferior corners of the cranial vertebra and anterior–superior corner of caudal vertebra onto the bisectrix. Lens distortion was corrected using a software program using the artificial intelligence algorithm with a ratio of 0.017232 and scale of 0.935914 in the pilot study. To compensate for variations in stature and radiographic magnification, the mean depth of the caudally adjacent vertebra was used to normalize the measurements of intervertebral alignment relationships.

### 2.5. Statistical Analysis

For the microcirculation status, the paired Student’s t-test was used to analyze the variation of the microcirculatory perfusion between the baseline and postintervention within each group. Independent Student’s t-test was used to compare the difference in the effects of intervention and videofluoroscopic imaging data between asymptomatic participants and patients with postural neck and shoulder pain. According to the central limit theorem, the distribution of sample means approximates a normal distribution regardless of whether the source population is normal or skewed, provided the sample size is sufficiently large (usually n > 30). As the microcirculatory perfusion was monitored on the participants bilaterally, there were collectively 32 and 44 measurements for asymptomatic and patient groups respectively in this study. Consequently, the F-test was performed to compare the standard deviations of the two groups under the promise of normal distribution. *p* < 0.05 was considered statistically significant. Analyses were performed using the Scientific Package for Social Sciences (SPSS for Windows, version 15.0; SPSS Inc., Chicago, IL, USA).

## 3. Results

We included 16 asymptomatic participants and 22 patients with postural neck and shoulder pain (mean age = 26.3 ± 2.4 and 25.4 ± 3.2 years, respectively; *p* > 0.05). The demographic information of asymptomatic participants and patients with postural neck and shoulder pain are presented in Table 1. The mean Neck Disability Index (NDI) of the asymptomatic participants was significantly lower than that of the patients with postural neck and shoulder pain (*p* < 0.001).

The microcirculatory blood flow is proportional to both the intensity and the Doppler shift of the light scattered back by the moving cells. The algorithm used to calculate the blood flow has been reported in our previous study [32]. Here, the microcirculatory flow measured through LDF is expressed using an arbitrary unit defined using a designated calibration process. The means and standard deviations of the baseline blood flow in the upper trapezius, SCM, and masseter muscles in both the groups are listed in Table 2. The difference in the baseline blood flow between the asymptomatic and patient groups was nonsignificant in the three muscle sites. However, the standard deviations in the upper trapezius muscle were significantly large (*p* < 0.05)—regardless of whether low-power or high-power LDF was used.

Compared with the individual baseline, the postintervention blood flow in the superficial skin (detected by low-power LDF) significantly increased at the SCM muscles and that in the deep skin (detected by high-power LDF) significantly increased at the upper trapezius, SCM, and masseter muscles in the asymptomatic participants (all *p* < 0.01). By contrast, both superficial and deep skin microcirculation increased significantly (all *p* < 0.01) at three sites after the intervention in patients with postural neck and shoulder pain.

Figure 2a illustrates the blood flow ratio of the asymptomatic participants and patients to compare the effects of massage intervention on microcirculations between groups. The blood flow ratio was higher in the patient group than in the asymptomatic group when it was monitored through low-power LDF at all muscle sites as well as when it was monitored through high-power LDF at the masseter and SCM muscles (Figure 2b). The *p* values in Figure 2 for the intergroup comparison in the three muscles are listed in Table 3.

The intraclass correlation coefficients (ICC) for calculating the intervertebral movement varied between 0.823 and 0.955 (average = 0.903), and the corresponding mean absolute difference (MAD) between examiners was 0.51°. The interexaminer ICC of the calculated disc spaces ranged from 0.824 to 0.926 (average = 0.875), and the corresponding MAD between examiners was 0.61°. Our results indicated that the high-reliability tests within and between examiners for identifying the vertebral landmarks and intervertebral movement in sequential videofluoroscopic images are high.

The intervertebral flexion and extension movements in the C2/3, C3/4, C4/5, C5/6, and C6/7 segments for the asymptomatic participants and the patients with postural neck and shoulder pain were shown in Figure 3. The total ROMs in the C2/3, C3/4, and C4/5 segments between the asymptomatic participants (35.05° ± 7.78°, 35.57° ± 8.46°, 29.27° ± 9.37°) and the patients with postural neck and shoulder pain (26.78 ± 5.97°, 27.58° ± 6.65°, 24.78° ± 7.14°) were significantly different (*p* = 0.005, 0.012, and 0.037, respectively).

## 4. Discussion

To provide additional insights into cervical biomechanics related to postural neck and shoulder pain and muscular intervention from the microcirculatory aspect, the resting and postintervention microcirculation in the major neck muscles were monitored. The results demonstrated that even though the difference in the averages of the resting (baseline) between the asymptomatic participants and the patients was nonsignificant, the significantly larger standard deviation revealed a larger variation of resting value among patients. Regarding the effects of the muscular intervention on skin microcirculation, the blood flow ratio of >1 in both the patient and asymptomatic groups, but the effects were more significant in the patients than the asymptomatic participants.

Compared with the asymptomatic group, the patient group has a larger standard deviation of the resting skin blood flow, which potentially revealed that microcirculatory regulation was impaired in some patients with neck and shoulder pain, especially around the upper trapezius muscle. The large standard deviation indicated that the data were more spread out—revealing that some patients were in a high-blood-flow state but the others were in a low-blood-flow state. In the study of the correlations of neck and shoulder skin microcirculation and pain symptoms in office workers with sedentary lifestyles [26], the skin microcirculation in the neck and shoulder regions was found to be significantly lower in the high-pain group than in the low-pain group. Our results suggested that the state of the lower resting blood flow could be related to microcirculatory insufficiency. By contrast, the symptom of the high blood flow was also found in Chia’s [27] study, where the microcirculatory perfusion was also monitored through high-power LDF. Chia found that the high-blood-flux phenomenon in the neck and shoulder regions was only found in adults with a sedentary lifestyle, whose neck and shoulder pain levels were rated as 2.8 ± 2.4 on a 10-point visual analog scale of 0 (*no pain*) to 10 (*unbearable pain*). However, this phenomenon was not found in individuals with regular physical activities, whose pain levels were only 1.5 ± 2.2. In addition, Strøm et al., suggested the importance of the interaction between microcirculatory vasodilation and nociceptors in the activation of muscle nociceptors [22] because the authors noted a positive correlation between pain and blood flux in people with chronic shoulder and neck pain. Prolonged unnatural body positions afford excessive strain on muscles or soft tissues and therefore lead to mild and unnoticeable muscle and fascia injury; here, we found that the high blood flow symptom may be the physiological response associated with chronic inflammation evoked by a mild injury. In Mork’s [33] investigation, a forward-bend posture and an increased trapezius muscle blood flow were found during computer work with visual (direct glare) stress. We proposed that the alterations in the posture could be a factor involved in the changes in the trapezius muscle activity. The observed increase in the trapezius blood flow could be influenced by changes in a more forward-bent posture affecting muscles around the neck and shoulder regions. In the clinical observation of the present study, some patients with neck and shoulder pain demonstrated a forward head posture compared with the healthy individuals. This possibly led to increased loading at the upper trapezius muscle, which has a greater moment arm to keep the weight of the head. We suggested that this was another possible reason that the state of high blood flow was found at the upper trapezius muscle in the patients.

Although the variation in baseline blood flow is so large that distinguishing whether the tissue is chronically injured by the value of blood flow is impossible, the larger massage-induced blood flow ratio was found in the patients with pain (Figure 2). Notably, transverse massage could elicit skin hyperemia (indicated by a blood flow ratio of >1). However, the superficial (epidermal) hyperemia reaction gradually faded (or even became nonsignificant) and became less obvious than that in the dermis in the asymptomatic participants. Nevertheless, the hyperemia reaction in the epidermis remained ongoing in the patients with pain and even the superficial reaction around the upper trapezius muscle was significantly higher than that in the deep skin (*p* < 0.01). The blood flow ratio was significantly larger in the patients than in the asymptomatic participants—regardless of whether low-power or high-power LDF was used for monitoring. Considering that the blood circulation system is similar to a water supply system, the end of the system will be the last to be irrigated; here, when the water supply decreases, water will retreat. As blood supplementation is not needed to repair injuries in healthy people, the circulation system regulates and makes the stimulating effect of massage smaller or fade away quickly. As low-power LDF measures the blood flow in the superficial layer (the end of the microcirculation), the hyperemia measured through low-power LDF was found to be less obvious, whereas the responses faded but remained significant in deep skin in the asymptomatic participants. By contrast, for the patients with pain, hyperemia due to massage can treat local injuries; therefore, hyperemia can be maintained for a long period not only in the deep skin but also in the epidermis.

Transverse friction massage (TFM) is commonly used for identifying muscle injuries, relieving pain, reducing adhesion, and improving circulation. Bell [17] applied a massage therapy program to effectively increase ROM, reduce overall spinal pain, and assist healing of tissues affected by disk herniation. TFM was found to be effective in reducing myofascial trigger point tenderness and increasing active ROM [19,20]. The higher hyperemia reaction induced by TFM among the patients with pain revealed that TFM could trigger an acute blood supplement similar to acute inflammation, which could facilitate the recovery of the local muscle injuries and therefore promote healing of neck and shoulder pain. Furthermore, as mentioned above, the present results revealed that the high blood flow symptom (i.e., chronic inflammation) induced in the patients with pain—which occurred in a constant state—appeared to be an essential injury-healing mechanism.

The phenomenon of hyperemia maintenance after massage (Figure 2) also appeared to be consistent with Røe’s [24] findings, where the intervention of massage was replaced with standardized data-terminal work with time and precision demands. Considerable hyperemia was observed in the upper trapezius muscle of the operating shoulder not only during work but also immediately after works on a computer—even though surface electromyography demonstrated no muscle activity after work. In Strøm’s [22] study, blood flux in the active upper trapezius muscle did not return to the baseline level during the initial 15 min of the recovery in patients with chronic should and neck pain in contrast to the immediate return to the baseline level in people without pain. A similar result was reported by Rosendal et al. [34]—where blood flow remained high in patients with chronic trapezius myalgia during the initial 20 min of recovery after a repetitive low-force exercise, but it decreased to baseline levels in the control individuals. Compared with the aforementioned studies using LDF with an invasive single fiber or microdialysis to monitor postintervention hyperemia, the postmassage hyperemia in the skin noted when noninvasive LDF was used was also found in the present study.

The differences in the total ROM in the C2/3, C3/4, and C4/5 segments between the asymptomatic participants and patients with postural neck and shoulder pain were significant. The SCM muscle was most active in the extended neck posture, whereas the upper trapezius muscle was most active in the flexed neck posture. This anatomical orientation may require the aforementioned muscles to support the shoulder during forceful arm exertion and lifting. A study found that patients with chronic neck pain demonstrate reduced motion and altered patterns of muscle control in the SCM and upper trapezius muscles during specific coordination tasks [35]. The authors found that workers with pain exhibited relatively high activity in the neck extensor muscles and inability to relax after the tasks. The current study indicated that compared with the asymptomatic participants, the patients with postural neck and shoulder pain had significantly lower intervertebral motion and muscular flexibility around the neck. Moreover, the NDI of the asymptomatic participants was significantly lower but that of the patients with postural neck and shoulder pain had significantly higher. Compared with the asymptomatic participants, the patients with postural neck and shoulder pain possibly had more irritative, tight muscles, and worse blood circulation.

As the massage therapies were effective in increasing active ROM and decreasing overall pain, the higher hyperemia reaction caused by TFM provides indirect evidence of improved microcirculation for the healing effect. This finding revealed that postintervention hyperemia determined through noninvasive LDF could be developed as an indicator for mild and unnoticeable muscle and fascia injuries due to an inappropriate or prolonged static posture. In addition, the patients’ high-blood-flow state should be considered for early treatment if it is related to chronic inflammation and occurs constantly. This is because chronic inflammation has been a well-established risk factor for cardiovascular diseases [36,37]. With the advantages of low radiation and real-time visualization of vertebral segments, videofluoroscopy could be promising, feasible, and highly reliable for clinical and research applications. The quantitative analysis of the intervertebral motion and muscular microcirculation may be employed to diagnose movement abnormalities like hypomobility or hypermobility and to monitor the treatment effect, and it may reveal more complicated or even compensatory movements for the spinal impairment. However, our study focuses only on the participants of similar symptoms and ages. Future researches should expand the subjects across different ages and spinal problems. Another limitation of this study is the sampling depth of LDF which is only to the dermis. To understand the phenomenon in muscles, instruments with deeper diagnosis depth should be applied in the future study. Furthermore, the significance of the regulatory mechanism underlying the acute hyperemia caused by muscular intervention still deserves more researches combined with other physiological characteristics, such as biochemical factors, to obtain a more complete understanding.

## 5. Conclusions

Our findings can serve as the basis for understanding neck and shoulder pain and the effects of muscular intervention from the microcirculatory aspect. Compared with the asymptomatic participants, the patients with neck and shoulder pain had higher NDI and limited cervical ROM. Simultaneously, the massage-induced responses at the skin microcirculation around the muscles controlling the head and neck posture and motion were considerably vigorous. In addition, from the microcirculatory measurement of resting state, the ischemic and hyperemic skin perfusions were only noted in the patients with pain. With the advantages of safety and non-invasive measurement, LDF could be applied for further research on musculoskeletal disorders with different muscular interventions for developing assessment methods and therapeutic strategies.

## Figures and Tables

**Figure 1 diagnostics-11-02226-f001:**
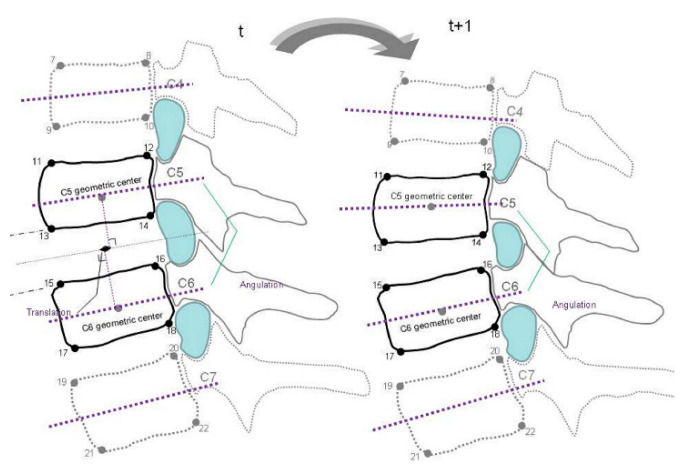
The methods for identifying vertebral landmarks.

**Figure 2 diagnostics-11-02226-f002:**
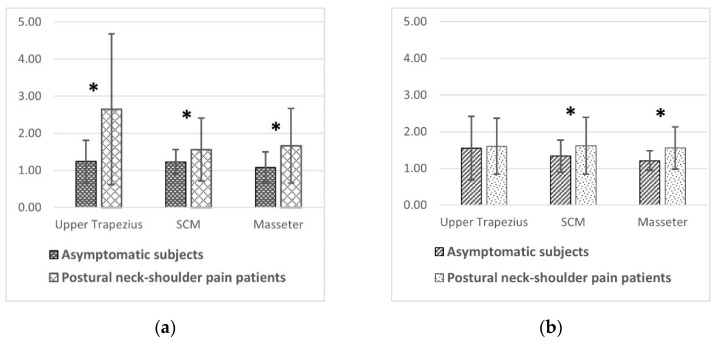
Mean blood flow ratio of the asymptomatic participants and patients with postural neck and shoulder pain in the upper trapezius, SCM, and masseter muscles across the transverse friction massage monitored by (**a**) low-power and (**b**) high-power LDF. The length of the error bar represents one standard deviation. *: *p*<0.05.

**Figure 3 diagnostics-11-02226-f003:**
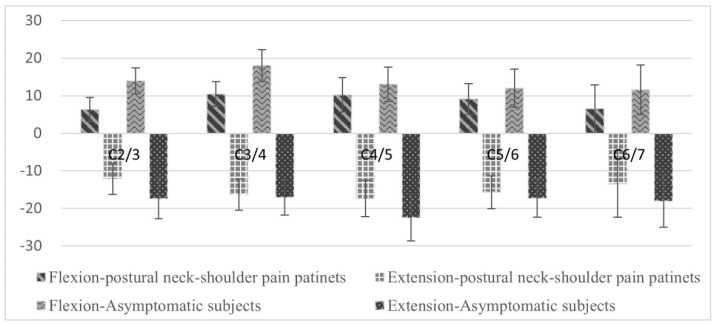
Mean intervertebral flexion and extension movements of the asymptomatic participants and patients with postural neck and shoulder pain. The length of the error bar represents one standard deviation.

**Table 1 diagnostics-11-02226-t001:** NDI values and demographic characteristics of asymptomatic participants and patients with postural neck and shoulder pain (means ± standard deviations).

	Asymptomatic Subjects (n = 16)	Postural Neck–Shoulder Pain Patients (n = 22)	*p* Value
Age	26.3 ± 2.4	25.4 ± 3.2	>0.05
Height	168.4 ± 8.7	166.7 ± 9.6	>0.05
Weight	65.2 ± 6.8	67.3 ± 7.7	>0.05
Neck Disability Index	5.98 ± 3.04	16.25 ± 2.95	*p* < 0.001

**Table 2 diagnostics-11-02226-t002:** Blood flow in the three muscles of asymptomatic participants and patients with postural neck and shoulder pain (means ± standard deviations).

Blood Flow		Low Power LDF			High Power LDF		
		Upper Trapezius	SCM	Masseter	Upper Trapezius	SCM	Masseter
Asymptomatic subjects	Mean ± SD	17.4 ± 8.3	20.5 ± 8.6	37.3 ± 17.7	93.6 ± 41.8	83.5 ± 28.5	122.2 ± 58.6
Postural neck-shoulder pain patients	Mean ± SD	17.4 ± 12.3	19.0 ± 10.2	32.6 ± 16.2	109.7 ± 72.0	78.9 ± 39.2	130.0 ± 66.8
*t*-test	*p* value	0.995	0.499	0.238	0.227	0.573	0.602
F-test	*p* value	0.048	0.676	0.690	0.029	0.361	0.700

**Table 3 diagnostics-11-02226-t003:** *p* values between asymptomatic participants and patients with postural neck and shoulder pain in the three muscles.

	Low Power LDF			High Power LDF		
	Upper Trapezius	SCM	Masseter	Upper Trapezius	SCM	Masseter
*p* value	<0.001	0.024	0.002	0.793	0.047	<0.001

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
