# Peer review of "Myofascial Treatment for Microcirculation in Patients with Postural Neck and Shoulder Pain"

_diagnostics, 2021, doi:10.3390/diagnostics11122226_

Round 1

Reviewer 1 Report

The study aims to characterize the microcirculation associated to neck and shoulder pain due to incorrect posture, and the effect of muscular intervention on microcirculation.

The authors performed a case-control study by comparing 16 healthy subjects with 22 age-matched patients suffering by shoulder and neck pain. Laser-doppler flowmetry (LDF) was used to characterize the difference in the microcirculation in the upper trapezius, sternoclenomastoid (SCM), and masseter muscles, and videofluoroscopic imaging to identify vertebral landmarks at the levels C2-C7. Results showed that patients group had higher standard deviation in blood flow. Patients underwent a treatment of Transverse Friction Massage (TFM) and and blood flow ratio after treatment was higher than healthy controls.

Abstract: in the abstract there is no mention to the use of videofluoroscopic imaging. Please add it

Introduction: the reported references seem quite outdated. Can the author find some more recent literature about the arguments of the paper?

Paragraph 75-89: the sentences in this paragraph seem disconnected. Please restate this paragraph to make it clearer

Line 76: maybe stating “cervical muscle dysfunction” would make the concept clearer

Line 77: please insert a reference for this statement

Line 85: please define TFM acronym and briefly describe in what this treatment consists

Line 126: please define here the acronym NDI. Please also add a reference for this scale

Line 156: what was the patient position?

Line 167: since data were obtained bilaterally, please add which data you used for analysis.

Line 175: Please add more details on acquisition protocol: which movements and how many repetitions? What was the patient position?

Line 190: please define which computer programs you used, if you used the software of the videofluoroscopy system or if you created custom functions with another software, and, in case, specify which software you did use

Lines 201-206: please add the significance level of the tests and the normality test. Please also describe here the F-test you performed and the analysis you performed on videofluoroscopic imaging data

235-237: please describe this analysis in the material and methods section

Figure 2: please add to the figure if data reported are the mean or the median and what the error bars stand for

Lines 246-248: please describe this analysis in the material and methods section

Lines 257-264: please remove the values if they are the same in figure and just report the significative ones

Figure 3: please add the error bars

Line 319: there is a reference to a figure not in the manuscript (Figure 4)

Discussion: please add the limits to the study and the further studies you may do to validate or elaborate on you results

Author Response

Please see the attachment for the responses. The modified parts were marked with red letters.

Reviewer 2 Report

Thank you for opportunity for reviewing this interesting paper. The research adhere to reporting CONSORT guidelines. This paper provides useful information on evaluate the effect of myofascial Treatment for microcirculatory blood flow in patients with postural shoulder and neck pain in Taiwan . I suggest to respond major concerns that may be addressed in order to clarify several considerations:

TITLE

The title of this manuscript is very long. Perhaps a more concise version for clarity, interes and ease of read.

ABSTRACT

It is hard to get the detail in an abstract when the word count is limited and this is often the hardest part of a paper to write. However, I do feel that it would be beneficial to explain what specifically you are looking at in relation to effect of myofascial Treatment for microcirculatory blood flow in patients with postural shoulder and neck pain (this also applies to the main body of the paper).  This needs to be made clearer throughout the paper

KEYWORDS: 

Please use recognised MeSH terms as this will assist others when they are searching for information on your research topic. The following website will provide these (simply start typing in a keyword and see if it exists or find an alternative if it does not): https://www.ncbi.nlm.nih.gov/mesh

INTRODUCTION

The introduction is weak and very short. An introduction should announce your topic, provide context and a rationale for your work, while catching the reader´s interest and attention. The above has not been given in the introduction that I have read.

Thus, I suggest in this section should be improved, with more details about prevalence, see researches of Martínez Jiménez et al  related with Acute effects of myofascial induction technique in plantar fascia complex in patients with myofascial pain syndrome on postural sway and plantar pressures https://pubmed.ncbi.nlm.nih.gov/32114316/. and Flexor Digitorum Brevis Muscle Dry Needling Changes Surface and Plantar Pressures: A Pre-Post Study https://pubmed.ncbi.nlm.nih.gov/33451013/

In addition, authors should be careful when making certain statements. In this regard, they affirm in the lines 40 to 44  that "(...) In addition to the static posture, the nature of the repetitive computer tasks may affect the movement patterns in the neck and shoulder regions. The cumulative effects of the working conditions in a limited space and with inappropriate workplace design, which demand a static posture, may contribute to an increased risk of musculoskeletal discomfort.

Also, please describe the hypothesis in this section.

MATERIAL AND METHODS: 

This section is poor, needs to present a better rationale for the study and the methodology employed. Also, neither appear information related with inclusion and exclusion criteria, dates, protocol.  The study design is a cuasi expermiental study, where this study was conducted in the hospital or in the outpatient center?. 

Also, Please, expand and clarification information related with the calculate sample size.

Lastly, please provide the number ethics committee of Medical Research Ethics Committees United or to explain  aspects ethics and legal requirement about this research.

RESULTS

The results need provide clear results and describe them. Please include the table 2   the p-values in all test in these tables

DISCUSSION: 

Include this section the principal strengths and weaknesses in relation to other studies, discussing important differences in results; the meaning of the study: possible explanations and implications and unanswered questions and future research

CONCLUSION:

These conclusions need to be softened, modified a in order to reflect only the study findings.

Author Response

Please see the attachment for the responses. The modified parts were marked with red letters in the revised manuscript.

Round 2

Reviewer 1 Report

The authors have amended the manuscript and it is suitable for publication in the present form

Reviewer 2 Report

I thank the corresponding author for their comments. I have read through the subsequent changes made to the manuscript and I have no further comments or suggestions.